# MULTILINGUAL CODE RETRIEVAL WITHOUT PAIRED DATA: NEW DATASETS AND BENCHMARKS

## ABSTRACT

We seek to overcome limitations to code retrieval quality posed by the scarcity of data containing pairs of code snippets and natural language queries in languages other than English. To do so, we introduce two new datasets. First, we make a new evaluation benchmark available, dubbed $M^2CRB$, containing pairs of text and code, for multiple natural and programming language pairs – namely: Spanish, Portuguese, German, and French, each paired with code snippets for: Python, Java, and JavaScript. The dataset is curated via an automated filtering pipeline from source files within GITHUB followed by human verification to ensure accurate language classification. Additionally, in order be able to train models and evaluate on the proposed task, we pose the following hypothesis: if a model can map from English to code, and from other natural languages to English, then the model can directly map from those non-English languages into code. We thus build a training corpus made of a new paired English/Code dataset we curate, and further combine it with existing translation datasets given by pairs of English and other natural languages. Extensive evaluations on both our new tasks as well as on existing code-to-code search benchmarks confirm our hypothesis: models are able to generalize to unseen language pairs they indirectly observed during training. We examine a broad set of model classes and report the influence of different design choices on the observed generalization capabilities.

## 1 INTRODUCTION

Recent work has demonstrated remarkable progress in settings where one's goal is to obtain code snippets conditional on natural language queries. In the generative setting for instance, code models such as ALPHACODE (Li et al., 2022b) obtained human-level performance when generating code from competitive programming problem statements posed in plain English, and several other successful examples merit mention, such as CODEX (Chen et al., 2021), CODEGEN (Nijkamp et al., 2022a), PALM (Chowdhery et al., 2022), STARCODER (Li et al., 2023), WIZARDCODER (Luo et al., 2023), and LLAMA (Touvron et al., 2023), to name but a few. Similarly, in the retrieval setting, CPT-CODE (Neelakantan et al., 2022) showed that contrastive training of encoders using pairs of docstring and code results in a semantic embedding where code search from text can be efficiently performed. Moreover, approaches such as CODEBERT (Feng et al., 2020) highlighted that representations extracted from BERT-like models (Devlin et al., 2018; Liu et al., 2019) trained on code succeed on code search upon fine-tuning on specific language pairs, and similarity scores such as CODEBERTSCORE (Zhou et al., 2023), a variation of BERTSCORE (Zhang et al., 2019) tailored for code, perform well as an automatic evaluation metric for generated code if fine-tuned on different combinations of English with specific programming languages.

As evidenced by the examples above, the quality of code retrieval and generation from natural language queries is rapidly increasing. However, there's a focus on using English as the underlying language of the source query, and multi-source-language models are still scarce. As a consequence, in contexts where developers do not speak English as a first language or are required to write non-English documentation, translation steps must be introduced. The focus on the use of English as a source language is partially due to the lack of large scale paired data mapping between different natural languages and code. While multilingual models were proposed, that has been the case on tasks where parallel data are not required such as causal language modeling (Scao et al., 2022). In contrast, multilingual text-to-code has only been partially addressed with English being paired with

different programming languages. Search approaches such as both CPT-CODE and CODEBERT for instance are evaluated on the CODESEARCHNET benchmark (Husain et al., 2019) where, given a query in English, the model retrieves a code snippet deemed relevant among 1000 candidates. Similarly, MBXP (Athiwaratkun et al., 2022) was built as a transpiled version of MBPP (Austin et al., 2021) into various programming languages, but source queries are still in English.

We address the limitations posed above and enable training of models to map multi-source-natural-language queries to multi-target-programming-language code. To do so, we first create an evaluation dataset that follows CODESEARCHNET in style, but contains queries in multiple natural languages. In possession of that, we carry out an extensive empirical analysis and determine efficient training approaches that result in multilingual models *without relying on intermediate translation steps*. That is, we show that combining multiple paired datasets suffices to enable mapping between languages only indirectly linked. *E.g.*, training to map Portuguese to English and English to Python enables directly mapping Portuguese into Python. Our analysis focuses on the *code search from text* task for two main reasons. First, it's a highly relevant setting in practical situations where a query must result in reliable and tested code from an existing codebase. Moreover, the search setting is less compute-intensive relative to common language models, rendering experimentation more accessible. Nonetheless, we remark that the data we introduce can be directly applied to evaluate text-conditional code models, and the training approaches we evaluate can be re-used in that setting as well.

Our contributions are summarized as follows:

1. We introduce a new *in-the-wild* evaluation dataset dubbed M$^2$CRB, where multiple natural languages are used to search over a codebase containing multiple programming languages. In particular, the data contains docstrings in Spanish, Portuguese, German, and French, all paired with code snippets in languages such as Python, Java, and JavaScript. The dataset is filtered from real-world data and thus *reflects the challenges of real practical applications*. (§ 2)

2. We substantially supplement the training partition of CODESEARCHNET with additional data for a subset of the programming languages they consider (English to Python, Go, Java, and JavaScript). In particular, the natural/programming languages observed in this complementary dataset are chosen to be such that they are never seen at testing time. (§ 3.2)

3. We contribute a training recipe that enables search over unseen language pairs. *I.e.*, we show that indirectly paired languages at training time can be directly mapped when testing. We report effects of different design choices and the performance of various fine-tuning schemes, and extensively evaluate a broad set of model classes and sizes, both in our new and other two existing datasets. (§ 3.1)

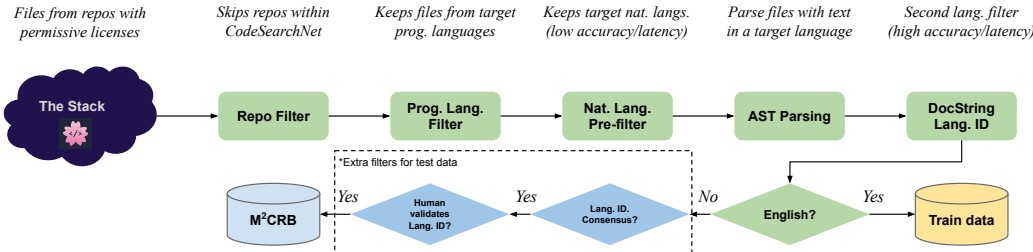

Figure 1: Training and testing data preparation workflow. A dataset is created with an automated workflow followed by human verification of resulting data. Note that non-English docstrings are verified by humans and samples that do no correspond to the predicted language are *discarded*.

## 2   MULTILINGUAL CODE RETRIEVAL BENCHMARK (M$^2$CRB)

We build a new multilingual text-to-code evaluation task with a mix of automated filtering and human verification. Specifically, we use data collected from THE STACK (Kocetkov et al., 2022), which contains over 6TB of permissively-licensed source code files from GITHUB. The workflow used to filter and prepare data is illustrated in Figure 1, and its steps can be summarized in the following:

    1. Filtering out repositories that appear in any split of CODESEARCHNET.

2. Keeping only the files that belong to the programming languages of interest.

3. Pre-filtering the files that likely contain text in the natural languages of interest.

4. AST parsing (we used TREE-SITTER[1] for that purpose).

5. An ensemble of classifiers performs language identification of docstrings.

6. Human verification of docstrings predicted as non-English by the entire ensemble with consensus. Humans filter out misclassifications from the language classification ensemble.

In further detail, programming language filtering is done via file extensions. At step 3 on the above, for each file, we perform what we call a *pre-filter operation* and check the fraction of overlaps between words and the union of vocabularies on the natural languages of interest. We keep only the files for which that fraction is greater than a threshold. Remaining files are AST parsed, and we finally perform language identification with three independent classifiers on each docstring, *i.e.*, we ensemble three different off-the-shelf open-source language classifiers from text[2,3,4]. The pre-filtering operation is done so as to avoid running the language identification ensemble on data that are likely to not contain text in languages of interest. If the ensemble majority voting for a docstring is English, that function/method is added to the training set (cf. § 3.2 for details). Otherwise, *and only if the three language predictors in the ensemble agree*, we perform a further step and ask a human to verify the ensemble's prediction. Fluent speakers of each of the considered natural languages verified each non-English docstring to ensure they correspond to the correct language. The resulting non-English functions are added to M[2]CRB, and the final row counts per natural/programming language pair are reported in Table 1.

| Natural Lang. | Programming Lang. | | | Total |
|---|---|---|---|---|
| | Python | Java | JavaScript | |
| Spanish | 1298 | 598 | 631 | 2527 |
| Portuguese | 1376 | 694 | 450 | 2520 |
| German | 543 | 928 | 83 | 1554 |
| French | 779 | 164 | 199 | 1142 |
| Total | 3996 | 2384 | 1363 | 7743 |

Table 1: Row count of M$^2$CRB per combination of the underlying natural language of docstrings and programming language of corresponding code. Preview and download data at: https://huggingface.co/datasets/blindsubmissions/M2CRB

| Natural Lang. | Programming Lang. | | | Avg. |
|---|---|---|---|---|
| | Python | Java | JavaScript | |
| Spanish | 0.836 | 0.742 | 0.737 | 0.772 |
| Portuguese | 0.823 | 0.743 | 0.737 | 0.768 |
| German | 0.813 | 0.701 | 0.721 | 0.745 |
| French | 0.827 | 0.723 | 0.722 | 0.757 |
| Avg. | 0.825 | 0.727 | 0.729 | 0.760 |

Table 2: Expected BERTSCORE measure between docstrings and code in M$^2$CRB.

In Table 2, we assess M$^2$CRB's docstring/code semantic alignment in terms of the expected BERTSCORE (Zhang et al., 2019; Zhou et al., 2023). In other words, we measure to what extent natural docstrings match the code they describe. To do so, we used the multilingual STARENCODER[5](Li et al., 2023), a BERT (Devlin et al., 2018) variation trained on multiple natural and programming languages. High scores are observed across all language combinations (BERTSCORE maps a pair of sequences onto $[-1, 1]$), especially so for Python as verified in results discussed in § 4.3.1 and § 4.3.2, and indicates that naturally occurring text/code pairs resulting from our filtering pipeline is well aligned to an extent that it can be considered for evaluation of text-to-code or code-to-text models such as retrievers and/or conditional generators. Further qualitative analysis using what we called GPTSCORE is reported in Table 7 where we used a GPT-3.5-TURBO as an evaluator of alignment between docstrings and code.

---

[1]https://tree-sitter.github.io/tree-sitter/
[2]https://github.com/saffsd/langid.py
[3]https://github.com/google/cld3
[4]https://huggingface.co/papluca/xlm-roberta-base-language-detection.
[5]https://huggingface.co/bigcode/starencoder

## 3 MULTILINGUAL TRAINING WITHOUT FULLY PAIRED DATA

In possession of the M²CRB evaluation data provided once the pipeline described above is executed, we studied approaches to enable generalization to its natural and programming language combinations assuming no paired training data for those language combinations would be available. We thus introduce a training strategy where we map multiple natural languages to English, and English to a number of programming languages. We finally show that to enable directly mapping non-English queries into code. In what follows, we provide a more detailed description of the setting we consider (§3.1) and how we combine different datasets to realize the training approach we propose (§3.2).

### 3.1 PROBLEM SETTING

Naive supervised training of multi-language models would require $|\mathcal{S}| \times |\mathcal{T}|$ parallel datasets, where $\mathcal{S}$ and $\mathcal{T}$ correspond to the sets of source natural and target programming languages, respectively. To work around that, we leverage a language-invariant semantic embedding, where encoded data depends only on the underlying implementation represented by programming languages or their descriptions represented in text. In doing that, we can then elect one of the source datasets as the anchor, denoted $S^* \in \mathcal{S}$, and instead train models using parallel datasets between sources and the anchor, and between the anchor and the targets. In other words, we replace the need for parallel data between all elements of $\mathcal{S}$ and $\mathcal{T}$ for parallel data between elements of $\mathcal{S}$, which enables the use of available natural language translation datasets. An illustration of the described training scheme is provided in Figure 2. In this example, $\mathcal{S} = \{\text{English}, \text{Portuguese}\}$, $\mathcal{T} = \{\text{Python}\}$, and $S^* = \text{English}$. At training time, models observe pairs of sentences in English and Python snippets as well as Portuguese and English pairs. At testing time, we then search Python directly from queries in Portuguese. Put simply, we test whether models generalize to new language combinations, only *indirectly* paired during training.

More generally, given the set $\mathcal{S}$ of source data distributions along with the set $\mathcal{T}$ of target data, training will require paired data from one of the sources, which will be referred to as *anchor* and denoted $S^*$, and all the targets. In addition, we further require access to paired data between the anchor and the remaining source domains. Our training dataset thus corresponds to the union of finite samples observed from the following dis-

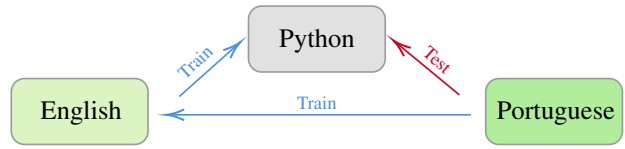

Figure 2: Illustration of the scheme we consider. *Hypothesis*: if a model is able to map from English (the anchor) to Python and from Portuguese to English, then it should be able to directly map from Portuguese to Python.

tributions: $(S^*, T) \, \forall \, T \in \mathcal{T}, (S^*, S) \, \forall \, S \in \mathcal{S} - S^*$, where the parenthesis notation $(\mathbb{P}, \mathbb{Q})$ indicates the joint over distributions $\mathbb{P}$ and $\mathbb{Q}$. For testing on the other hand, we seek models able to map between any combination of source and target distributions. Our test sets are then sampled from the union of the full set of joints: $(S, T) \, \forall \, S \in \mathcal{S}, T \in \mathcal{T}$. Note that any data realization $x$ observed from any distribution $x \sim \mathcal{D} \in \mathcal{S} \, \cup \, \mathcal{T}$ indicates a sequence with length $L$ of symbols or tokens from a shared vocabulary $\mathbb{V}$: $x = [x_1, ..., x_L] : x_i \in \mathbb{V}$. Moreover, we highlight that we use the terms *map/mapping* in a rather general sense to mean either a generative case where one literally translates from source to target, or a discriminative case where one retrieves from a set of candidate targets given an instance from the source. As will be further discussed in § 4.1, both types of mappings may be used when training models, but only the discriminative one is considered during testing.

### 3.2 HOW TO BUILD TRAINING DATA INDIRECTLY PAIRING LANGUAGES

We combine a mix of both existing paired datasets of different kinds and new data we introduced to build a dataset with the properties described in § 3.1. We set the anchor $S^*$ to English, and $\mathcal{T}$ to the set of programming languages represented in CODESEARCHNET, *i.e.*, $\mathcal{T} = \{\text{Python}, \text{Go}, \text{Java}, \text{JavaScript}, \text{PHP}, \text{Ruby}\}$. The pairs represented by $(S^*, T) \, \forall \, T \in \mathcal{T}$ are given by the training partition of CODESEARCHNET (Husain et al., 2019) considerably augmented with data we prepared, *i.e.*, train data with docstrings in English as per Fig. 1. To incorporate anchor/-source pairs, we leveraged machine translation tasks. Namely, a subset of WMT-19 (Wikimedia-Foundation, 2019) was considered given by its English/German and English/Finnish partitions. We

additionally included non-anchor source/source combinations to increase the amount of training data. As such, we used the French/German subset of WMT-19 as well as the Spanish/Portuguese and Spanish/Galician partitions of TATOEBA (Tiedemann, 2020). A single target/target case is considered and is given by the code translation data within CODEXGLUE (Lu et al., 2021). Summary statistics of the complete training data are shown in Table 3 including our contributed GH-X.

| Dataset | Row count | Sampling proportion ($\times 10^{-5}$) | Epochs |
|---|---|---|---|
| CODESEARCHNET | 1880853 | 0.0532 | 3.5 |
| GH-Python (*Ours*) | 15000002 | 0.0067 | 0.5 |
| GH-Java (*Ours*) | 15000014 | 0.0067 | 0.5 |
| GH-GO (*Ours*) | 15000078 | 0.0067 | 0.5 |
| GH-JavaScript (*Ours*) | 2000040 | 0.0500 | 3.3 |
| WMT-19 (DE-EN) | 1995208 | 0.0501 | 3.3 |
| WMT-19 (FR-DE) | 1999990 | 0.0500 | 3.3 |
| WMT-19 (FI-EN) | 2000000 | 0.0500 | 3.3 |
| TATOEBA (ES-PT) | 67777 | 1.4754 | 29.1 |
| TATOEBA (ES-GL) | 3132 | 31.9285 | 30.0 |
| CODEXGLUE | 10300 | 9.7087 | 30.0 |

Table 3: Statistics of each of the datasets used for fine-tuning. Given the variability in size of each dataset as indicated by the row counts, we adjust sampling proportions during training so that datasets are uniformly represented in the actual training sample. The number of epochs represented in the rightmost column indicates the approximate number of times we iterate over the entire dataset each time we feed approximately 30 billion tokens to a model. GH-X, the data we introduced, is hosted at: https://huggingface.co/datasets/blindsubmissions/GH_text2code.

## 4 EXPERIMENTS

### 4.1 MODELS AND TRAINING

We leverage models that were pre-trained on CODESEARCHNET, and fine-tune them in the training data mix summarized in Table 3. We covered a number of different methods and models classes such that encoder-only, dual encoders, encoder-decoder, and decoder-only models were all considered. Moreover, we covered the 60M-360M parameters range, which is rather typical for sequence-level representation learning. Namely, we fine-tune popular models such as CODET5 (Wang et al., 2021), CODEBERT (Feng et al., 2020), and CODEGEN (Nijkamp et al., 2022a), and further consider dual-encoder models (Karpukhin et al., 2020). Besides those models we fine-tuned ourselves, we also evaluated state-of-the-art embeddings not trained by us (cf. Table 9 with results from OPENAI's and SENTENCE TRANSFORMERS's embeddings). However, we highlight that we have no control over training of these embeddings and fairness of comparison is unclear. Those results were then kept in the appendix to serve as evidence that one can indeed predict from M²CRB to good accuracy.

Training varies depending on the architecture being fine-tuned, and while all models we consider train against a contrastive objective, models able to generate train in a multitask setting where conditional causal language modeling is also employed. We thus evaluate models both on our new dataset and on other two existing benchmarks to address questions such as:

1. Are models able to generalize to language combinations only indirectly paired during training?

2. What is the effect in retrieval performance given by different approaches to allocating parameters (*i.e.,* encoder-decoder vs. dual-encoder at a fixed parameter budget)?

3. Can generalization be expected between target languages to perform code-to-code tasks?

Our main training objective is a contrastive loss that depends on encoding input sequences into vectors denoted $z_{EMB}$, and such vectors are encouraged to match for corresponding source and target instances. For example, docstrings and corresponding implementations should map to similar vectors in terms of some distance measure. Similarly, matching sentences in two different natural languages should embed into neighboring points, as should two code snippets implementing the same

functionality but written in different programming languages. Encoders, denoted $\mathcal{E} : \mathcal{V}^L \mapsto \mathbb{R}^D$ for a $D$-dimensional embedding, will be implemented as the output at a special token (*e.g.*, [EOS]) appended to all inputs. Given a batch of $n$ pairs of sequences $[x_i, y_i]_{i=1}^n : x_i, y_i \sim (S \in \mathcal{S}, T \in \mathcal{T})$, our contrastive objective relies on the similarity matrix $Sim$ where each entry $Sim[i, j] \in [0, 1]$ is the rescaled cosine similarity measured between embeddings of $x_i$ and $y_j$:

$$Sim[i, j] = (1 + \cos(\mathcal{E}(x_i), \mathcal{E}(y_j)))/2. \tag{1}$$

We then force $Sim$ to be an identity matrix. That is, the objective $\mathcal{L}_{contrastive}$ encourages that the similarity between paired data will be greater than that between unpaired instances, *i.e.*:

$$\mathcal{L}_{contrastive} = ||Sim - I_n||_2. \tag{2}$$

Similar to CLIP (Radford et al., 2021) however, we implement a variation of $\mathcal{L}_{contrastive}$ that shares its minimizers since it's easier to train against this variation, as observed empirically. The loss we use treats $Sim$ as a batch of logits, and places labels on the main diagonal to define a cross-entropy objective. An implementation of $\mathcal{L}_{contrastive}$ is shown in Figure 15.

For models able to generate, we use an additional autoregressive maximum likelihood objective:

$$\mathcal{L}_{generative} = \frac{1}{n} \sum_{i=1}^n \prod_{t=1}^{L'} p_\mathcal{D}(y_i^t | y_i^1, ..., y_i^{t-1}, \mathcal{E}(x_i)). \tag{3}$$

We then take advantage of the fact that all training datasets are parallel, and define translation auxiliary tasks. In cases where $\mathcal{L}_{generative}$ can be obtained, we further introduce a denoising objective in which the original sequence is to be recovered from its noisy version where tokens are randomly dropped out, and the dropping out probability is assumed to be a tunable hyperparameter.

Given the two types of losses mentioned above, we then train with their convex combination:

$$\mathcal{L} = \alpha \mathcal{L}_{contrastive} + (1 - \alpha) \mathcal{L}_{generative}, \tag{4}$$

where $\alpha \in [0, 1]$ is a hyperparameter weighing the importances of the two components.

As for pre-processing, subword tokenization is performed following the same strategy as in the pre-training of each model we fine-tune, *e.g.*, byte-pair encoding (Sennrich et al., 2015) for CODET5. Moreover, given a training pair $(x, y)$, we append special tokens and use the following template: $\{lang[y]\} : \{x\}[EOS]\{y\}[EOS]$, where the operator $lang(\cdot)$ returns the underlying language of its argument. Data augmentation is further performed during training so as to avoid spurious solutions, able to retrieve by simply matching keywords present in both docstrings and actual implementations. In early experiments, we noticed well performing models would simply retrieve code snippets for which variable or function names would appear in the docstrings. To counter that and enforce actually semantic retrieval, we remove this shortcut by randomly assigning meaningless function names in codes snippets, and via randomly replacing variable names by uninformative strings.

## 4.2 EVALUATION METRIC

Given paired sets of $n$ points from the source and target distributions as denoted by $X, Y = x_1, ..., x_n, y_1, ..., y_n \overset{\text{i.i.d.}}{\sim} (\bigcup_{i=1}^{|\mathcal{S}|} \mathcal{S}_i, \bigcup_{i=1}^{|\mathcal{T}|} \mathcal{T}_i)$, evaluations will require computation of the mean reciprocal rank defined by:

$$\text{MRR}(X, Y) = \frac{1}{n} \sum_{i=1}^n \frac{1}{\text{rank}(x_i, y_i, Y)}, \tag{5}$$

where the rank corresponds to the position/index of $y_i$ in the ordered set of similarities:

$$\text{rank}(x_i, y_i, Y) = \text{index}(Sim(x_i, y_i), \{Sim(x_i, Y)\}_{sorted}), \tag{6}$$

and $\text{index}(\cdot, \{\cdot\})$ returns the position of its first argument in the ordered set given as the second argument. Note that we overload $Sim$, defined in the same way as that used to compute entries of the similarity matrix in (1), and compute it both for data pairs, where the output is a scalar, and between a source data point and a set of target domain instances, in which case a set of similarities is output. We then evaluate models under varying sizes of the underlying sets. That is, we compute the area under the MRR curve, obtained for increasing sizes of retrieval sets, as indicated in the following:

$$au\text{MRR}c(X, Y) = \int_0^1 \text{MRR}(X, Y_{1:\lfloor n * t \rfloor}) dt. \tag{7}$$

To approximate (7), we discretize $t$ so that $t = \{5\%, 10\%, 20\%, 30\%, 50\%, 75\%, 100\%\}$.

| | | \textsc{CodeT5} | | | | \textsc{CodeBERT} | \textsc{CodeGen} |
|---|---|---|---|---|---|---|---|
| | | *Small* | *Base* | *Base-Encoder* | *Base-Dual-Encoder* | | |
| *Python* | de | 60.2% | 63.4% | 58.1% | 73.2% | 91.8% | 65.7% |
| | es | 74.8% | 70.0% | 75.5% | 89.1% | 93.3% | 63.8% |
| | fr | 70.3% | 68.1% | 64.5% | 78.8% | 93.6% | 68.2% |
| | pt | 66.9% | 62.7% | 70.6% | 85.9% | 94.0% | 51.0% |
| *Java* | de | 23.2% | 26.5% | 31.3% | 19.2% | 27.6% | 21.7% |
| | es | 27.3% | 31.8% | 37.2% | 18.2% | 25.6% | 22.0% |
| | fr | 39.3% | 44.3% | 47.4% | 34.4% | 46.3% | 49.9% |
| | pt | 26.4% | 31.1% | 34.8% | 23.6% | 28.1% | 26.6% |
| *JavaScript* | de | 62.2% | 56.1% | 61.6% | 56.9% | 56.9% | 55.7% |
| | es | 22.2% | 25.1% | 28.6% | 18.4% | 23.3% | 19.6% |
| | fr | 30.5% | 30.1% | 30.8% | 27.2% | 32.6% | 28.0% |
| | pt | 20.8% | 22.2% | 27.2% | 16.6% | 20.0% | 14.3% |
| *Avg.* | | 43.7% | 44.3% | 47.3% | 45.1% | 52.7% | 40.5% |

Table 4: *au*MRR$c$ (the higher the better) on M$^2$CRB, *i.e.*, code search from natural language queries in different languages. All models are fine-tuned as described in §4.1 and further detailed in Appendix D. Note that non-fine-tuned models yield an *au*MRR$c$ close to zero.

## 4.3 RESULTS AND DISCUSSION

Evaluations are split into three parts where the first two evaluate to what extent models manage to generalize to unseen language combinations. We finally carry out an in-distribution evaluation on English-to-code tasks. Variations of \textsc{CodeT5}[6] are considered such as different model sizes, *i.e.*, SMALL ($60M$ params.) and BASE ($200M$ params.), training using only its encoder, and a dual-encoder variation, indicated as DUAL-ENCODER in tables and figures, where each encoder only observes either natural or programming language. We further consider \textsc{CodeBERT}[7] ($124M$ params.) and \textsc{CodeGen}[8] ($350M$ params.). *All models start from their pre-training weights publicly available, and we fine-tune following the procedure described in* § 4.1. That is, whenever possible, a multitask approach uses both contrastive representation learning and causal language modeling to train. On the other hand, models that are only able to encode input data are trained against the contrastive objective only. Further training details as well as hyperparameters and information regarding model sizes can be found in Appendix D along with extra results from general-purpose embeddings.

### 4.3.1 MULTI-LANGUAGE EVALUATION WITH M$^2$CRB

We perform evaluations on M$^2$CRB in terms of *au*MRR$c$ as reported in Table 4 for different fine-tuned models, while MRR curves as a function of $t$ are displayed in Figure 7 in Appendix C. In particular, we are concerned with assessing to what extent models manage to generalize to source/target pairs unseen during training. We further seek to compare fully discriminative encoder-only models with multitask models able to both retrieve and generate.

We first note that all models we fine-tuned are able to generalize to language pairs unobserved during training. Non-fine-tuned models are as good as a random embedding in this task, with *au*MRR$c$ close to zero. Contrary to what is usually observed in more standard evaluation conditions (Kaplan et al., 2020), in this multilingual setting, scaling up model size will not necessarily improve performance. More importantly, adding a decoder won't hurt performance significantly, and it can be done if the generation capability is desired. That is, if generation is important downstream, using multitask models will not hurt retrieval performance in the multi-source-language case to a very large extent. It also suggests that retrieval-augmented language models (Borgeaud et al., 2022; Zhou et al., 2022; Adlakha et al., 2022) can be trained without an external pre-trained retriever.

---

[6] https://huggingface.co/Salesforce/codet5-base
[7] https://huggingface.co/microsoft/codebert-base
[8] https://huggingface.co/Salesforce/codegen-350M-multi

| | CODET5 | | | CODEBERT | CODEGEN |
|---|---|---|---|---|---|
| *Small* | *Base* | *Base-Encoder* | *Base-Dual-Encoder* | | |
| 66.4% | 80.0% | 81.4% | 50.2% | 44.4% | 19.0% |

Table 5: *au*MRR*c* for code search from code queries on the Python-Java data of (Roziere et al., 2020). All models are fine-tuned (cf. § 4.1) and non-fine-tuned models's *au*MRR*c* is close to zero.

| | CODET5 | | | CODEBERT | CODEGEN |
|---|---|---|---|---|---|
| | *Small* | *Base* | *Base-Encoder* | *Base-Dual-Encoder* | | |
| *PHP* | 33.9% | 31.7% | 33.2% | 39.3% | 43.8% | 39.0% |
| *JavaScript* | 39.5% | 39.1% | 40.4% | 38.6% | 45.3% | 41.0% |
| *Python* | 71.4% | 64.4% | 65.6% | 89.5% | 89.7% | 67.2% |
| *Go* | 61.6% | 56.8% | 61.0% | 73.8% | 75.5% | 61.1% |
| *Java* | 36.3% | 32.7% | 37.6% | 36.2% | 39.3% | 36.4% |
| *Ruby* | 37.0% | 37.7% | 48.2% | 47.1% | 53.5% | 47.9% |
| *Avg.* | 46.6% | 43.7% | 47.7% | 54.1% | 57.9% | 48.8% |

Table 6: *au*MRR*c* for code search from English for the test set of CODESEARCHNET. All models are fine-tuned (cf. § 4.1). Non-fine-tuned models are omitted since their *au*MRR*c* is close to zero.

We further highlight that adding a second encoder won't improve performance to a great extent. This is not surprising considering that each encoder in dual-encoder models trains on only half the data as compared to a single encoder. Aligned with this observation, it has been argued in previous work that training data diversity influences performance on testing conditions that differ from training (Albuquerque et al., 2019). Finally, the decoder-only model doesn't reach the performance level of alternatives, which indicates that the strategy of concentrating the full parameter budget in a causal decoder underperforms encoder-only models in retrieval, even under a model size advantage.

### 4.3.2 EXTRA EVALUATION WITH EXISTING DATA: CODE-TO-CODE RETRIEVAL

We push evaluated models further and test their ability to generalize to unseen pairs of target/target language combinations. To do so, we use the Python-Java paired data collected from GEEKSFORGEEKS and discussed in (Roziere et al., 2020) given by implementations in both languages of the solution of a given problem. In other words, models are tasked with querying a codebase in Java using Python snippets. During training however, natural-programming language or natural-natural language combinations make it for the bulk of the training data. Results are reported in Table 5 while MRR curves are displayed in Figure 6 for language combinations unseen to models during training. Once more, results indicate that models are able to generalize to unseen combinations of domains, however the gap between encoder-only and the multitask encoder-decoder models grows significantly now that tasks shifted to programming-programming language combinations. Also, the decoder-only and dual-encoder models observed a much bigger gap relative to the other cases in this scenario. Removing capacity or training data from the encoder seems particularly detrimental to retrieval performance when the task shifts more relative to training. Note that the gap between the CODET5's encoder and CODEBERT suggest that different pre-training strategies influence downstream performance significantly since the gap is reversed w.r.t. the text-to-code evaluations (cf. Tables 4 and 6).

### 4.3.3 EXTRA EVALUATION WITH EXISTING DATA: CODE SEARCH FROM QUERIES IN ENGLISH

Finally, we run a more standard English to code evaluation using the test set of CODESEARCHNET to assess the in-distribution effect of including extra language combinations in the training set. Results are in Table 6 and MRR curves are displayed in Figure 8. Similarly to the multi-source language evaluation discussion in Section 4.3.1, we did not observe too large a gap between encoder-only and encoder-decoder models, suggesting once more that evaluation tasks where source/target pairs are closer to what was observed during training will result in a smaller gap between the two types of models. However, contrary to the other evaluations, adding a second encoder significantly improves performance relative to the encoder-only counterpart, and CODEBERT was the best performer on

average in this case. The decoder-only model also observed a more favourable scenario in this particular evaluation. This shift in results highlights that in-distribution performance will not always correlate with performance under unseen testing conditions, and evaluation on multilingual data such as M$^2$CRB is required in order to reliably assess multilingual capabilities. Note that results for Python are consistently better relative to other languages, as anticipated by scores in Tables 2 and 7.

## 5 RELATED WORK

The success of transformers for natural language modelling (Vaswani et al., 2017) has motivated their use for code. CODEPARROT (Tunstall et al., 2022) was trained with data from GITHUB for code completion on a single language. CODEGEN (Nijkamp et al., 2022b), POLYCODER (Xu et al., 2022), CODEX (Chen et al., 2021), and CODET5 (Wang et al., 2021) were trained on additional programming languages as well as natural language queries in English. Here, we fine-tune pre-trained code models to support unseen additional natural and programming language combinations. Close to our fine-tuning approach, recent work has leveraged contrastive techniques to learn a useful sentence- or document-level embedding where retrieval can be performed efficiently. GTR (Ni et al., 2021) for instance, showed that dual-encoder settings can benefit from scaling up model size. SIMCSE, on the other hand, showed that simply dropping out tokens leads to effective augmentation approaches to create positive pairs for unsupervised sentence representation learning. In settings involving embedding both text and code, DOCCODER (Zhou et al., 2022) uses a contrastive scheme to match representations from natural language queries to retrieve documentation later used for code generation. Most similar to models we evaluate, CPT-CODE (Neelakantan et al., 2022) was pre-trained with contrastive learning to learn a common space between text in English and code. Similarly, CODERETRIEVER (Li et al., 2022a) combines contrastive objectives using an unsupervised approach to determine similar function pairs. In both cases however, multilinguality is not tackled, nor is code-to-code retrieval, which we show to be attainable in this contribution.

Acquiring aligned code and natural language is key to solve tasks such as code retrieval and summarization. Towards this goal, Yin et al. (2018) proposed CONALA, mined from STACKOVERFLOW containing English paired with Python and Java. MCONALA (Wang et al., 2022) extended it to provide English, Spanish, Japanese, and Russian text to Python pairs. Similarly, other datasets were built with data from online forums which makes it so that natural languages we consider are not well represented (Yao et al., 2018; Hu et al., 2020), besides being based on Q&A pairs rather than having text queries that describe a code snippet. Closer to our proposal, CODESEARCHNET (Husain et al., 2019) leveraged English comments from GITHUB repositories. We extend CODESEARCHNET by a factor of approximately 25, and add a new multilingual test set. An attempt towards defining multi-programming-language translation systems was carried out in TRANSCODER (Roziere et al., 2020; 2021) where multiple unparalleled data sources are considered. Beyond that, we address the question of whether models can map multiple natural languages to multiple programming languages.

## 6 CONCLUSION

We introduced M$^2$CRB, a new evaluation dataset where paired data are available containing source and target data given by multiple natural languages and programming languages. Crucially, since M$^2$CRB comprises naturally occurring text and code from GITHUB, it then defines an *in-the-wild* set of tasks, akin to a search application over a real codebase. Enabled by these testing data, an extensive empirical evaluation in the code search/retrieval setting was carried out in order to indicate how different design choices influence performance under an out-of-distribution generalization condition we consider, given by new language combinations presented to models at testing time. We showed that one can overcome the lack of multi-language paired training data by introducing indirect paths from source to target languages. Interestingly, we verified that one can dedicate capacity to a decoder to enable generative capabilities without affecting retrieval performance significantly. Also, results showed that once tasks diverge more from training, then removing either capacity or training data from encoders is detrimental to performance. We further observed that in-distribution evaluation is not a good indicator of performance on new language pairs, and evaluation data such as M$^2$CRB comprising language combinations of interest is necessary. For future work, while we cover 16 scarce combinations of programming and natural languages, M$^2$CRB can be expanded to further improve its coverage especially so in terms of additional programming languages.

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

| Natural Lang. | Programming Lang. | | | AVG. |
|---|---|---|---|---|
| | *Python* | *Java* | *JavaScript* | |
| *Spanish* | 0.80 | 0.62 | 0.38 | 0.60 |
| *Portuguese* | 0.86 | 0.54 | 0.36 | 0.59 |
| *German* | 0.80 | 0.16 | 0.30 | 0.42 |
| *French* | 0.94 | 0.36 | 0.28 | 0.53 |
| **AVG.** | 0.85 | 0.42 | 0.33 | 0.53 |

Table 7: GPTSCORE measured between docstrings and code in M$^2$CRB.

## A    DATA EVALUATION WITH GPTSCORE

In Table 7, M$^2$CRB's qualitative analysis is performed in terms of a metric we refer to as GPTSCORE. To compute that metric, we used a random sample of size 50 for each language combination and prompted GPT-3.5-TURBO so that it would assess to what extent a docstring would be descriptive of a code snippet. The model then generates 1 for good pairs and 0 otherwise. Each entry in the table thus corresponds to the expectation of the model generations, and estimates the probability of such an evaluator approving a docstring as a good description of the provided code snippet. The exact prompt we used is shown in Figure 3.

We note that results align with what was observed in terms of BERTSCORE as reported in Table 2, and indicate variability in terms of how aligned naturally occurring text is with respect to the code it is supposed to document. Indeed, a relatively high alignment variance is not surprising given how large a community of developers GITHUB represents. The existence and quality of documentation is strongly dependent on common practices of sub-communities of developers, and hence strongly dependent on both the underlying natural and programming languages. On the other hand, natural data are reflective of what a *real-world* codebase would represent, and collecting data from open repositories defines a useful *in-the-wild* testbed.

```python
def make_question_prompt(docstring: str, function_str: str) -> str:
    request_str = f"I'd like for you to act as an evaluator of natural
        language descriptions of code snippets. Given as inputs a pair
        with the form:\ncode: 'some code'\ndescription: 'some
        description'\nI'd like for you to output 1 if you think that the
        description clearly explains the piece of code that was given. If
        you think otherwise, reply with a 0. Note that the answer must
        have exactly one character, and it should be 1 for good code
        descriptions and 0 for not so good descriptions of the code given
        as input. Here's the example for you to evaluate:\ncode:
        {function_str}\ndescription: {docstring}. What do you think about
        the goodness of that description in exactly one character, 0 for
        bad or 1 for good?"
    return request_str
```

Figure 3: Python function to make GPTSCORE's prompt.

## B    MODEL AND TRAINING DETAILS

Training configuration along with model details such as parameter counts are reported in Table 8. All models we evaluated are fine-tuned on the data mix detailed in Table 3. Fine-tuning was performed using Adam with a cosine schedule for the learning rate. In-domain cross-validation was performed in the validation partition of CODESEARCHNET to select hyperparameters, and batch sizes are selected to maximize accelerator hardware's memory usage. All models were trained on a single node containing 8 A100 NVIDIA GPUs.

The embedding procedure used to map input sentences/code snippets is illustrated in Figures 4 and 5 for encoder-decoder (or encoder-only) and decoder-only models, respectively. In all cases, we define

embeddings as the output at a particular special token so that contratsive training pushes models to output a sequence-level representation of past tokens once such special token is observed.

| | CODET5-SMALL | CODET5-BASE | CODET5-BASE-ENCODER | CODET5-BASE-DUAL-ENCODER | CODEBERT | CODEGEN |
|---|---|---|---|---|---|---|
| Param. Count | $61M$ | $224M$ | $110M$ | $220M$ | $124M$ | $359M$ |
| Base L.R. | 0.00001 | 0.00003 | 0.00002 | 0.0001 | 0.0001 | 0.00001 |
| Opt. Betas | 0.9, 0.999 | 0.9, 0.98 | 0.8, 0.999 | 0.9, 0.95 | 0.9, 0.95 | 0.9, 0.999 |
| Max. Grad. Norm | 1.0 | 1.0 | 1.0 | 1.0 | 1.0 | 1.0 |
| L2 Coeff. | 0.005 | 0.0001 | 0.0001 | 0.04 | 0.001 | 0.005 |
| Masking Prob. | 0.3 | 0.3 | 0.0 | 0.0 | 0.0 | 0.0 |
| Warmup epochs | 20.0 | 5.0 | 5.0 | 5.0 | 5.0 | 20.0 |
| Initial Temp. | 10.0 | 10.0 | 10.0 | 10.0 | 10.0 | 10.0 |
| Batch Size | 10.0 | 6.0 | 12.0 | 24.0 | 36.0 | 3.0 |
| $\alpha$ | 0.5 | 0.5 | 0.0 | 0.0 | 0.0 | 0.5 |

Table 8: Fine-tuning hyperparameters and model information for the models considered in our evaluations.

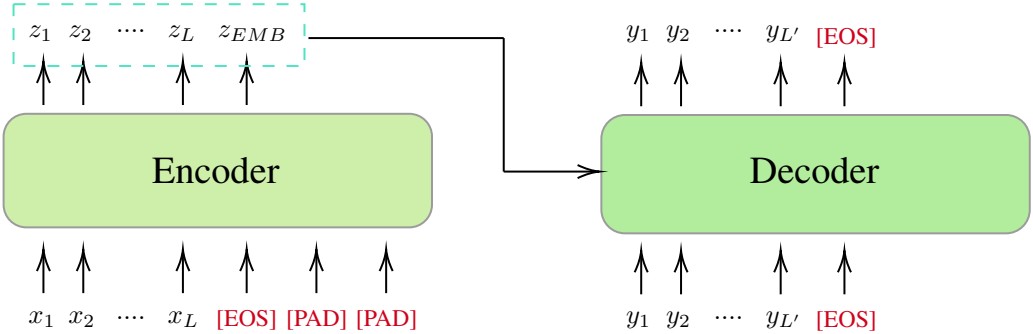

Figure 4: Encoder-decoder pair we consider in our experiments. Contrastive training is performed on top of embeddings obtained at the [EOS] token output by the encoder. Generative tasks, on the other hand, are performed with standard sequence-to-sequence maximum likelihood estimation.

## C  ADDITIONAL RESULTS

### C.1  $M^2$CRB EVALUATION WITH EXTERNAL EMBEDDINGS

In Table 9, we report additional results on $M^2$CRB when external embedding models are used. We remark that we have absolutely no control in terms of what models are used to generate external embeddings, and how these models were trained, rendering any comparisons likely unfair. We thus remark that results in Table 9 are not intended as to offer a comparison across different models. Rather, it serves as a means to give insight in terms of how available embedding services behave in these language combinations. In particular, we compared against open source embedding models from SENTENCE TRANSFORMERS[9] and closed source embedding from OPENAI's API[10]. Results from the CODEBERT model we fine-tuned are further included in the table for reference.

### C.2  $M^2$CRB EVALUATION WITH AUXILIARY TRANSLATOR

In Table 10, we report additional results on $M^2$CRB when an external translator is used. That is, given a textual query, we first translate it into English using some external service, and then query the model using the result. In particular, we used the public Google translate API[11] and compared our models with a model trained with only English to code data. We further report side by side comparisons of our models when we directly translate in the original language versus the cases where we translate beforehand. Both the encoder-only and encoder-decoder models perform better on average than a model trained exclusively on English to code, and that's the case no matter whether

---

[9] https://huggingface.co/sentence-transformers
[10] https://platform.openai.com/docs/guides/embeddings
[11] https://pypi.org/project/googletrans/

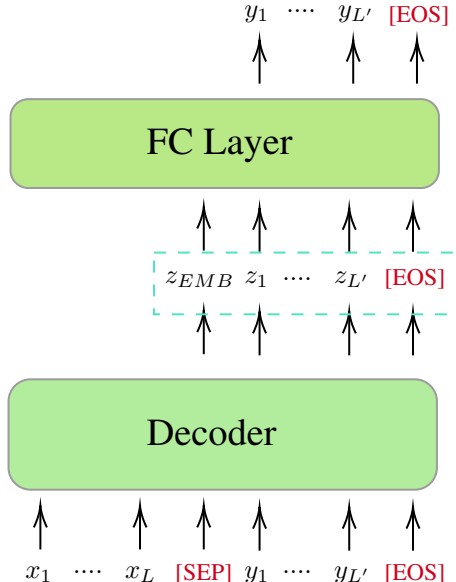

Figure 5: Decoder-only setting we consider in our experiments. Contrastive training is performed on top of embeddings obtained at the separator token [SEP] prior to final fully-connected layer. Generative tasks, on the other hand, are performed with standard sequence-to-sequence maximum likelihood estimation.

| | | Ours | Open source (SENTENCE TRANSFORMERS) | | Closed source (OPENAI) |
|---|---|---|---|---|---|
| | | CODEBERT | ALL-MINILM-L6-V2 | ALL-MINILM-L12-V2 | ADA-002 |
| Python | de | 91.8% | 67.7% | 73.3% | 92.9% |
| | es | 93.3% | 68.7% | 75.1% | 92.8% |
| | fr | 93.6% | 78.7% | 82.5% | 94.0% |
| | pt | 94.0% | 65.4% | 71.1% | 93.5% |
| Java | de | 27.6% | 16.0% | 25.6% | 43.6% |
| | es | 25.6% | 26.2% | 36.7% | 65.8% |
| | fr | 46.3% | 37.5% | 46.9% | 64.0% |
| | pt | 28.1% | 24.6% | 31.6% | 64.3% |
| JavaScript | de | 56.9% | 63.1% | 64.3% | 69.8% |
| | es | 23.3% | 26.0% | 32.5% | 59.2% |
| | fr | 32.6% | 38.3% | 43.2% | 59.2% |
| | pt | 20.0% | 25.1% | 30.4% | 49.5% |
| Avg. | | 52.7% | 44.8% | 51.1% | 70.7% |

Table 9: M$^2$CRB's results with external embeddings. Results correspond to the area under the MRR curve (auMRRc, the higher the better). Note that we have no control over external models and the trained data they employed for training. The comparisons in this table are likely unfair in that training data of external models might have included the test data.

| | | CODET5-ENCODER | | CODET5 | | CODET5 (En.) |
|---|---|---|---|---|---|---|
| | | *Direct* | *Translate* | *Direct* | *Translate* | |
| *Python* | de | 58.1% | 63.4% | 60.2% | 65.0% | 54.2% |
| | es | 75.5% | 69.4% | 74.8% | 66.9% | 60.2% |
| | fr | 64.5% | 64.5% | 70.3% | 66.1% | 57.2% |
| | pt | 70.6% | 63.5% | 66.9% | 61.6% | 54.7% |
| *Java* | de | 61.6% | 36.8% | 62.2% | 28.5% | 25.5% |
| | es | 28.6% | 39.1% | 22.2% | 37.0% | 34.5% |
| | fr | 30.8% | 55.4% | 30.5% | 45.9% | 48.8% |
| | pt | 27.2% | 40.6% | 20.8% | 35.0% | 33.7% |
| *JavaScript* | de | 31.3% | 60.4% | 23.2% | 61.9% | 52.2% |
| | es | 37.2% | 30.8% | 27.3% | 28.5% | 22.9% |
| | fr | 47.4% | 45.5% | 39.3% | 40.8% | 37.3% |
| | pt | 34.8% | 35.0% | 26.4% | 30.6% | 23.8% |
| *Avg.* | | 47.3% | 50.4% | 43.7% | 47.3% | 42.1% |

Table 10: M$^2$CRB's results with an auxiliary translator. Results correspond to the area under the MRR curve (*au*MRR*c*, the higher the better).

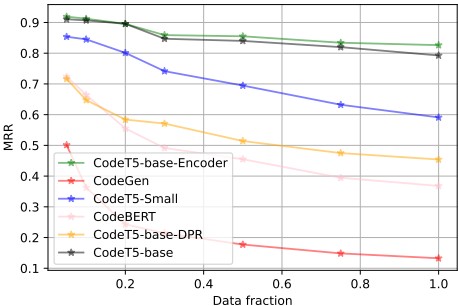

Figure 6: MRR curves for code to code evaluation.

or not an auxiliary translator is used. However, both our models benefited from the extra translation component on average, though that's not the case for every language combination.

## C.3 MRR CURVES

In the following, we complement results presented in Tables 4 (Fig. 7), 5 (Fig. 6), and 6 (Fig. 8) and plot MRR curves for various coverage levels of the search set.

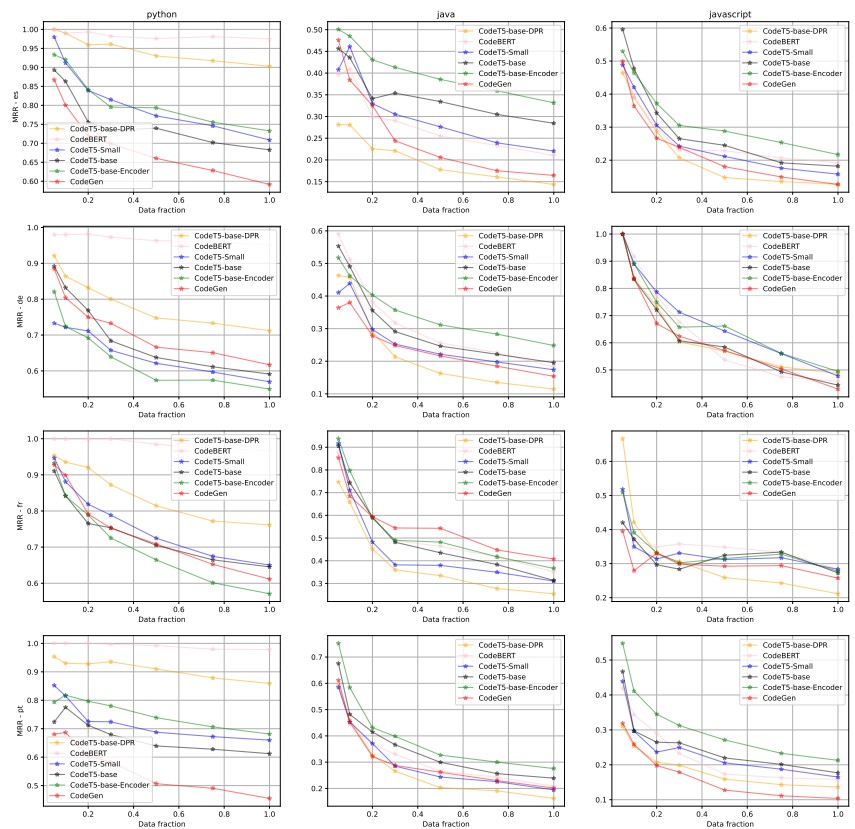

Figure 7: MRR curves for cross natural language evaluation.

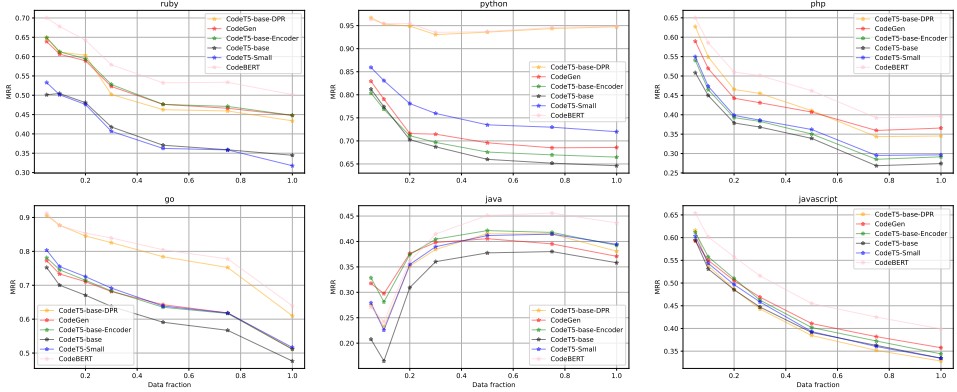

Figure 8: MRR curves for English to code evaluation.

## C.4 Data samples

M$^2$CRB can be previewed in the browser in https://huggingface.co/datasets/blindsubmissions/M2CRB, and download instructions are also provided in that same page. We further provide samples of various language combinations herein in Figures 9 to 14.

```python
def is_prop_symbol(s):
  """Um simbolo de logica de
      proposicao e uma string
      inicial maiuscula."""
  return is_symbol(s) and
      s[0].isupper()
```

Figure 9: Python and Portuguese.

```python
def exp_date(self):
  """Returne la date
      d'expiration de l'accred"""
  return self.renewal_date +
      datetime.timedelta(days=365)
```

Figure 10: Python and French.

```java
// Buscar producto por codigo id
    y devuelve el producto
    buscado completo
public Producto findById(String
    codigo) {
  int i = 0;
  boolean encontrado = false;
  //Mientras no hayamos llegado
      al final o encontrado lo
      que buscamos repetimos el
      bucle
  //Al encotrarlo, el bucle para
  while (i < lista.length &&
      !encontrado) {
    Producto deLista = lista[i];
    if (deLista.getCodigo()
    .equalsIgnoreCase(codigo))
      encontrado = true;
    else
      i++;
    }

  if (encontrado)
    return lista[i];//Devolvemos
        el producto buscado
  else
    return null;
}
```

Figure 11: Java and Spanish.

```java
// Es wird getestet ob eine
    Fehlermeldung ausgegeben
    wird, wenn bei der Funktion:
    len die laenge der
    Konstanten mit der Laenge
    des Textes uebereinstimmt.
public void setUpdated(String
    table)
    {
      provider.setUpdated(table,
          handle);
    }
```

Figure 12: Java and German.

## D  Additional training details

An implementation of the exact variation of $\mathcal{L}_{contrastive}$, similar to CLIP (Radford et al., 2021), is shown in Figure 15. In particular, it shares its minimizers with the loss described in the text, but was observed to be easier to train against in practice. The loss treats the similarity matrix $Sim$ as a batch of categorical log-probabilities, places labels on the main diagonal, and then defines a cross-entropy objective so that entries in the main diagonal are forced to be greater than off-diagonal elements.

Finally, we remark that we found that setting $\alpha = 0.5$ would work well for the encoder-decoder models we consider so that both generative and discriminative objectives are given the same importance. The same was applied for the decoder-only case.

```
\\ validacao de usuario por meio
    de middleware
function checkAllFields(body) {
 var keys = Object.keys(body);

 for (var _i = 0, _keys = keys;
    _i < _keys.length; _i++) {
  var key = _keys[_i];

  if (req.body[key] == "") {
    return {
      user: body,
      error: 'Por favor,
          preencha todos os
          campos.'
    };
  }
 }
}
```

Figure 13: JavaScript and Portuguese.

```
\\ donde estamos en el sistema-L
function setup() {
 createCanvas(710, 400);
 background(255);
 stroke(0, 0, 0, 255);

 // inicializar la posicion x e
    y en la esquina inferior
    izquierda
 x = 0;
 y = height - 1;

 // CALCULAR EL SISTEMA-L
 for (let i = 0; i < numloops;
    i++) {
  thestring =
      lindenmayer(thestring);
 }
}
```

Figure 14: JavaScript and Spanish.

```python
import torch

def contrastive_loss(
    x_source: torch.FloatTensor,
    y_target: torch.FloatTensor) -> torch.FloatTensor:
    """Computes contrastive loss.

    Args:
      x_source (torch.FloatTensor): Batch of normalized source
          embeddings. Expected shape is [batch_size, embedding_dim].
      y_target (torch.FloatTensor): Batch of normalized source
          embeddings. Expected shape is [batch_size, embedding_dim].

    Returns:
      torch.FloatTensor: Contrastive loss.
    """

    # Compute similarity matrix.
    sim = x_source @ y_target.T

    # A row-wise cross-entropy criterion is used
    # with labels placed on the main diagonal.
    ce_labels = torch.arange(sim.size(0)).long()
    contrastive_loss = torch.nn.functional.cross_entropy(
        sim,
        ce_labels)

    return contrastive_loss
```

Figure 15: Pytorch implementation of the contrastive loss we consider taking as inputs normalized representations of source/target data.

