# OpenReview forum: "Multilingual Code Retrieval Without Paired Data: New Datasets and Benchmarks"
_ICLR.cc/2024/Conference — Submitted to ICLR 2024_

### Official Review · Reviewer_mHNB · 2023-10-29

**Soundness:** 2 fair
**Presentation:** 2 fair
**Contribution:** 3 good
**Rating:** 6
**Confidence:** 3

**Summary:**

The paper studies the problem of multi-language code search, from different natural languages to different coding languages, or between different coding languages. A main contribution is a new evaluation set M^2CRB, which consist of en/pt/es/fr/de queries to python/java/javascript code snippets, all from real world GitHub repositories. The paper also brings up a hypothesis that the model should be able to learn the relationship between a language and a code with no direct training pairs, but bridged by an anchor language. English is used in this study.

Extensive experiments are done with different languages (natural and code), different sizes of CodeT5 models and different model architecture (CodeT5 with single/dual encoder, CodeBERT and CodeGen). Results show that the model can indeed learn to retrieve non-directly linked pairs of natural and coding languages. The effect of model architecture and sizes are also discussed.

**Strengths:**

- Introduced a new multilingual natural language to code evaluation. We need more realistic evaluation sets.
- Evaluated different models, sizes and languages extensively.
- Provided details of how the dataset is created.

**Weaknesses:**

Experiment about the effectiveness of data is not clearly setup.
- The purpose of adding TATOEBA and CodeXGlue is not clearly articulated. They also make the claim of learning through anchor languages weaker because another factor is added to the training mix.
- It's not sure if the pre-training data of various models already have multi-lingual exposure. The near-0 auMMR number could also be task alignment issues.
- Some ablation studies will help a lot to understand the role of each dataset.

Another weakness is in writing. It's common to find long or poorly written sentences that's hard to read, or not understandable at all to the Reviewer with moderate effort. See questions for some examples.

**Questions:**

- What's the purpose of the AST parsing phase?
- Table 2: "high scores are observed across all language combinations". It's better to have a baseline to help understand a relative assessment of "high".
- Section 3.1 last paragraph: "Note that any data realization x observed..." It's not clear why we need to discuss it here. Better to clarify the purpose.
- Table 3: Why do we need the TATOEBA/CodeXGlue set? What purpose do they serve in answering the research questions?
- Section 4.1 second paragraph: ".., models able to generate train ..." Reviewer couldn't understand what does this mean with moderate effort.
- Section 4.1 right after Equation 3: "... and define translation auxiliary tasks." What does this sentences mean?
- Section 4.3.2 last sentence: "Finally, the decoder-only model doesn't reach..." it's better to call out the model name (is it CodeGen?) to save the readers some back-n-forth work.
- Would you open source the data and evaluation code?

---

> ### Author Response · Authors · 2023-11-16
>
> We thank the reviewer for the very detailed review and for providing suggestions to improve our manuscript. We took those into great consideration and carefully responded to comments and questions in what follows.
>
> **1. Experiment about the effectiveness of data is not clearly setup.**
>
> **Response:** Assuming that the reviewer’s concerns are related to experiments with results reported in tables 2 and 7 (*kindly let us know if not, and we will address it*), we clarify that those correspond to estimates of the semantic alignment between text and code for each language combination. We did that by estimating the expected `BERTScore` between text and code pairs, and did that independently for each language combination. We did that using an encoder that was trained on multiple natural and programming languages.
>
> We also reported what we called the `GPTScore` which uses a similarity obtained from prompting a language model (GPT-3.5-turbo in this case) with a text-code pair. More details about this approach including the exact prompt we used are placed in the appendix due to space constraints.
>
> We observed both metrics to be predictive of retrieval performance, *e.g*., they suggest high semantic similarity for python and low for javascript, and those are the languages for which models consistently perform the best and the worst, respectively.
>
> **2. The purpose of adding TATOEBA and CodeXGlue is not clearly articulated. They also make the claim of learning through anchor languages weaker because another factor is added to the training mix.**
>
> **Response:** We used a relatively small number of source-source and target-target instances in order to make our training mix more diverse, and then help our models cope with shifts at evaluation (cf. [1] for a discussion on how diverse training data sources enable generalization to new, shifting distributions). CodeXGlue for instance gives us some code-code pairs. Even though the underlying languages differ from those we evaluate on, it’s a more similar task to those we expect at testing than the ones defined by other datasets.
>
> **3. It's not sure if the pre-training data of various models already have multi-lingual exposure. The near-0 auMMR number could also be task alignment issues.**
>
> **Response:** We completely agree with the reviewer. While base models were trained on multiple languages, they were never trained to yield chunk/sentence level embeddings that semantically align. Fine-tuning is thus required to align with test tasks of interest.
>
> **4. Clarity**
>
> **Response:** We thank the reviewer for kindly remarking points of improvements in terms of clarity and presentation of our contributions. We addressed each related comment in what follows.
>
> **5. What's the purpose of the AST parsing phase?**
>
> **Response:** We used AST parsing to be able to collect natural language from code snippets (docstrings and comments). We also followed CodeSearchNet and split files into functions/methods.
>
> **6. Table 2: "high scores are observed across all language combinations". It's better to have a baseline to help understand a relative assessment of "high".**
>
> **Response:** Thank you for pointing this out. We modified that sentence to make precise and clear what we mean by high similarity in that case.
>
> We clarify that all of those scores are bounded in $\[0,1\]$, so that sentence suggests that the scores are much closer to the upper limit than to the lower one. We further remark that a random embedding or random data would score very close to 0, and would not yield a very informative baseline. In other words, our goal in that sentence is to highlight that the scores suggest a reasonable signal-to-noise ratio in the data. This is further confirmed in our experiments in Section 4.3.1 and in table 9 in the appendix: the data can be predicted to high accuracy so the semantic similarity must be high.
>
>
> **7. Section 3.1 last paragraph: "Note that any data realization x observed..." It's not clear why we need to discuss it here. Better to clarify the purpose.**
>
> **Response:** In that paragraph, we are highlighting that when we use $x$ to refer to a data instance, we are referring to a sequence of tokens. This paragraph is intended to introduce some notation to make the manuscript accessible to readers maybe not so familiar with sequence modeling settings.
>
> **8. Section 4.1 second paragraph: ".., models able to generate train ..." Reviewer couldn't understand what does this mean with moderate effort.**
>
> **Response:** *”Models able to generate”* refer to models that can be used to generate text. In more detail, our evaluation covers encoder-only, encoder-decoder, and decoder-only architectures. As such, encoder-decoder and decoder-only both can be used for text generation, in addition to yielding chunk/sentence level embeddings.

---

> ### Author Response · Authors · 2023-11-16
>
> **9. Section 4.1 right after Equation 3: "... and define translation auxiliary tasks." What does this sentences mean?**
>
> **Response:** This sentence explains that the models that are able to generate text, *i.e.*, encoder-decoder and decoder-only models, are trained with an additional “generative” objective compared to contrastive-only encoders.
>
> To make the presentation clear, we will rephrase it and add an explicit reference to the generative objective discussed in Section 4.1 - Eq. (3).
>
> **10. Section 4.3.2 last sentence: "Finally, the decoder-only model doesn't reach..." it's better to call out the model name (is it CodeGen?) to save the readers some back-n-forth work.**
>
> **Response:** Yes, we agree with the reviewer and replaced it in the text.
>
> **11. Would you open source the data and evaluation code?**
>
> **Response:** Yes! The datasets are already (anonymously) available:
>
> - https://huggingface.co/datasets/blindsubmissions/M2CRB
> - https://huggingface.co/datasets/blindsubmissions/GH_text2code
>
> Links are also in the manuscript in the captions of tables 1 and 3 for training and testing data, respectively.
>
> Code for data processing, fine-tuning, and evaluation will be released upon publication.
>
> **References**
>
> [1] Albuquerque I, Monteiro J, Darvishi M, Falk TH, Mitliagkas I. Generalizing to unseen domains via distribution matching. arXiv preprint arXiv:1911.00804. 2019 Nov 3.

---

> > ### Author Response · Authors · 2023-11-19
> > **Checking for need of further discussion**
> >
> > We thank the reviewer for their positive assessment and for providing detailed feedback which helped improve our paper and make it more readable. We hope to have clarified the reviewer's concerns, but will be happy to follow-up with further discussion if necessary.

---

> > > ### Comment · Reviewer_mHNB · 2023-11-22
> > > **Thanks for the answers.**
> > >
> > > Thanks for the detailed answers.
> > >
> > > Regarding the introduction of two datasets (TATOEBA and CodeXGlue). Now I know what's the purpose, but it's still not clear that if this is necessary by some evidence in this particular study, or just an educated guess to try to make the final result better.
> > >
> > > I mostly maintain my current rating on the basis of supporting open source dataset and evaluation work.

---

> > > > ### Author Response · Authors · 2023-11-22
> > > >
> > > > Thank you for checking our responses and for replying. Indeed, our methodology set a fixed training dataset and the selection of data sources was not part of the ablations we carried out. The reason for that is simply a scoping one. We tried to answer the question "*can contrastive encoders search over language combinations only indirectly paired paired during training?*", but other possibly interesting questions such as "*What's the minimum number of language combinations required to induce that property?*" were not part of our scope.

---

### Official Review · Reviewer_QVKG · 2023-10-31

**Soundness:** 3 good
**Presentation:** 3 good
**Contribution:** 3 good
**Rating:** 6
**Confidence:** 3

**Summary:**

- This work introduces two datasets:
  - a Multilingual `(code snippet, NL queries)` paired data in 4 natural languages: Spanish, Portuguese, German, and French, and three programming languages: Java, JavaScript, and Python. The dataset is collected from GitHub data, with AST parsing and natural language identification, and then human validation to check for correct language identification. This dataset contains ~7700 data points and is used as a test dataset.
  - And, an `(English, code snippets)` paired data similar to the CodeSearchNet data
- This work then proposes a training setup to train code search models for languages for whom parallel data is not available. The training setup requires parallel data between one language ($S$) and code segments ($T$), and parallel data between all other languages ($S'$) and $S$. The model is then trained with a contrastive loss aimed at learning similar embeddings for similar code segments/natural languages and a standard autoregressive loss.
- This work then presents results for code search for the 4 languages introduced in the work, code search using English, and code-code search between Java and Python.

**Strengths:**

- The Code-NL Paired dataset between languages other than English has not been explored in prior works and could be useful for non-English speaking developers.
- From the statistics of the introduced data, it seems that, unlike English, data in these languages is not sufficiently available to train a model in the standard way. Thus, the authors propose to utilize available NL-paired datasets to indirectly learn code search for these new languages.
- This training setup, in future work, might also be explored to extend to code generation given NL descriptions in non-English languages. Additionally, the dataset proposed in this work could be used for evaluation in that setting as well.
- The paper is well-written and relatively easy to follow.

**Weaknesses:**

- There are no baselines presented to understand how well the proposed technique actually works. While baselines might be difficult to get for non-English code search (Table 4), I would assume for Python-Java code search (Table 5) and English code search (Table 6) available model embeddings should work well. For instance, CodeBERT reports an average MRR of 0.7603. It is not immediately clear what the auMRRc would be for this model, and it would have been helpful to get these numbers. Similarly for the Python-Java code search, it would have been nice to get baselines from available multi-lingual pre-trained models.
- This work requires paired NL data (such as between Spanish and English), and it incorporates this paired data in the loss function. However, another way to utilize this data could be to learn a translation model from English to Spanish using the paired data and use this translation model to translate English queries in the CodeSearchNet data to create a much larger (Spanish, Code snippet) dataset, albeit with some noise. This has the advantage of creating larger synthetic training data that can be directly used to train the code search model, instead of the objective proposed in the paper. Do the authors have some assessment on why the proposed technique is a better approach than this one?

**Questions:**

mentioned above

---

> ### Author Response · Authors · 2023-11-16
>
> We thank the reviewer for their comments and for the detailed assessment of the strengths and weaknesses they identified. We carefully address the latter below.
>
> **1.There are no baselines presented to understand how well the proposed technique actually works. While baselines might be difficult to get for non-English code search (Table 4), I would assume for Python-Java code search (Table 5) and English code search (Table 6) available model embeddings should work well. For instance, CodeBERT…**
>
> **Response:** We kindly highlight that we did evaluate other models not trained by us such as CodeBERT and CodeT5. We observed however that those models have no predictive power in the settings we consider and perform as well as a random embedding. Note that CodeBERT MRR results openly available typically correspond to the performance after fine-tuning in a particular language combination. **We went beyond that and evaluated a single fine-tuned model, trained in the multilingual scheme we proposed.**
>
> We further refer the reviewer to table 9 for results with state-of-the-art embeddings (OpenAI’s and Sentence Transformers’s) from models not trained by us, both open and closed source. Note that those results were kept in the appendix because we have no control over the training of the evaluated models, and in particular have no control over the data used for training. Still, those results attest for the predictability of the M$^2$CRB data and offer a performance upper bound that follow-up methods should strive to achieve.
>
>
> **2. This work requires paired NL data (such as between Spanish and English), and it incorporates this paired data in the loss function. However, another way to utilize this data could be to learn a translation model from English to Spanish using the paired data and use this translation model to translate English queries in the CodeSearchNet data to create a much larger (Spanish, Code snippet) dataset, albeit with some noise. This has the advantage of creating larger synthetic training data that can be directly used to train the code search model, instead of the objective proposed in the paper. Do the authors have some assessment on why the proposed technique is a better approach than this one?**
>
> **Response:** Thank you for commenting on the use of translation to generate training data. While we agree with the reviewer and, starting from an English-to-code dataset, one can indeed create synthetic variations of it by translating docstrings to different languages, it must be noted that translation step would add significant cost to pre-processing, and its quality could not be good enough considering we would be translating docstrings rather than natural language, and that we could be targeting low-resource languages.
>
> Cost and quality concerns aside, the proposal is indeed valid and orthogonal/complementary to what we show rather than a competing approach: if one does what the reviewer suggested and, for instance, translates docstrings in CodeSearchNet to Spanish and French, we showed that training a model in such a dataset would enable mapping directly between any combination of natural or programming languages. Either approach used to build a set of paired datasets would be equally good.
>
> In other words, combining existing training datasets or synthesizing via translation are both valid and complementary approaches to build a dataset with the properties our training requires. We highlight once more: our goal is to show that generalization is possible to indirectly observed language combinations.
>
> We added a sentence in Section 3.2 remarking that other approaches are possible to create the style of training data we require.

---

> > ### Author Response · Authors · 2023-11-19
> > **Checking for additional concerns**
> >
> > We thank the reviewer for their overall positive assessment of our contributions. We hope to have clarified the reviewer's concerns, but will be happy to follow-up with further discussion.

---

### Official Review · Reviewer_z3xc · 2023-11-01

**Soundness:** 3 good
**Presentation:** 3 good
**Contribution:** 2 fair
**Rating:** 5
**Confidence:** 4

**Summary:**

The paper introduces two new datasets to address code retrieval limitations due to the scarcity of data containing pairs of code snippets and natural language queries in languages other than English.
The first dataset, M²CRB, is an evaluation benchmark with text and code pairs for multiple natural and programming language pairs. The authors propose a training hypothesis that models can map from non-English languages into code if they can map from English to code and other natural languages to English. They create a training corpus combining a new paired English/Code dataset with existing translation datasets. Extensive evaluations confirm that models can generalize to unseen language pairs they indirectly observed during training. The paper contributes a new evaluation benchmark, additional training data, a training recipe for unseen language pairs, and an analysis of different design choices and fine-tuning schemes.

**Strengths:**

1. Introducing new datasets, M²CRB and the paired English/Code dataset, addressing the scarcity of multilingual code retrieval data.
2. Rigorous evaluation of various model classes and sizes on both new and existing datasets, confirming the proposed training hypothesis.
3. Clear presentation of the filtering pipeline, training methodology, and results, making it easy to follow and understand.
4. The study contributes to the understanding of multilingual code retrieval and offers a path for future work on more diverse language combinations.

**Weaknesses:**

1. While the M²CRB dataset covers multiple language pairs, it can be expanded to include more programming languages for better representation.
2. The study focuses on the code search/retrieval setting, but it would be helpful to investigate the applicability of the introduced data and training approaches in generative settings as well.
3. The evaluation focuses on models within the 60M-360M parameter range, and exploring larger-scale models could provide insights into the effect of model size on generalization capabilities in this domain.

**Questions:**

1. Can the training approach proposed in this paper be adapted for generative models, and if so, how would it affect their performance on text-to-code generation tasks?
2. Are there any potential biases in the dataset, such as the influence of specific programming language styles or the quality of non-English docstrings, that may affect the model's generalization capability?
3. How do the models perform when fine-tuned on different programming languages (e.g., C++, Rust, etc.) and less common natural languages? Would the performance be consistent with the results presented in the paper?
4. How would the results change when using larger-scale models, such as GPT-3 or the recent Megatron-LM? Would the generalization capabilities improve with increased model capacity?

---

> ### Author Response · Authors · 2023-11-16
>
> We thank the reviewer for the insightful review. We addressed all comments and questions in detail in the following.
>
> **1. While the M$^2$CRB dataset covers multiple language pairs, it can be expanded to include more programming languages for better representation.**
>
> **Response:** We agree with the reviewer. Kindly refer to the *Update plan* section of our dataset card in the following link where this is listed: https://huggingface.co/datasets/blindsubmissions/M2CRB
>
> **2. The study focuses on the code search/retrieval setting, but it would be helpful to investigate the applicability of the introduced data and training approaches in generative settings as well.**
>
> **Response:** We completely agree with the reviewer and also mentioned this in the text. Especially for the case of encoder-decoder architectures that require paired training data, the approach we proposed could enable multi-linguality efficiently for languages not necessarily combined during training.
>
> However, we decided to focus on the retrieval setting given its high practical importance, especially now that retrieval augmented generation (RAG) approaches become the prevalent language modeling framework.
>
> **3. The evaluation focuses on models within the 60M-360M parameter range, and exploring larger-scale models could provide insights into the effect of model size on generalization capabilities in this domain.**
>
> **Response:** We completely agree with the reviewer and highlight that the decision on the model scale we focused on was in part influenced by the amount of compute we had available. We also note that recent literature (*e.g.*, [1]) has shown that encoders do not observe scaling gains as pronounced as decoders. For instance, the extremely popular `sentence-transformers/all-MiniLM-L12-v2` encoder has only 33M parameters. Another recently released encoder, `bigcode/starencoder`, has 125M parameters. **We thus believe the range we chose to be very well aligned with common practice in this setting**.
>
> **4. Can the training approach proposed in this paper be adapted for generative models, and if so, how would it affect their performance on text-to-code generation tasks?**
>
> **Response:** As discussed above and in the manuscript, our proposed training strategy would be useful as-is to enable multi-linguality in encoder-decoder architectures. Even more than that, our results show that this can be done while also carrying out contrastive training on top of chunk-level embeddings obtained from the encoder. So to answer your question, **yes**, conditional generative models can benefit from our proposal, and **no**, we do not anticipate it affecting the performance of the base model it builds upon.
>
> **5. Are there any potential biases in the dataset, such as the influence of specific programming language styles or the quality of non-English docstrings, that may affect the model's generalization capability?**
>
> **Response:** Thank you for this insightful question. We measured language-specific text-code alignment (via `BERTScore` and our proposed `GPTScore`) and did observe variation across languages. Indeed, the textual component of the data we make available consists of naturally occurring comments on GitHub and, as such, relatively high variance is expected given how large a community of developers GitHub represents, and subcommunities will have different practices. On the other hand, natural data is reflective of what a real-world codebase would look like, and thus defines a useful "in-the-wild" testbed.
>
> **6. How do the models perform when fine-tuned on different programming languages (e.g., C++, Rust, etc.) and less common natural languages? Would the performance be consistent with the results presented in the paper?**
>
> **Response:** Our results suggest strong generalization ability to new language combinations for various model classes and training settings. We do not anticipate any issue specific to the languages highlighted by the reviewer, provided that they are included in the training mix (not necessarily paired as we’ve shown).
>
> **7. How would the results change when using larger-scale models, such as GPT-3 or the recent Megatron-LM? Would the generalization capabilities improve with increased model capacity?**
>
> **Response:** We kindly refer the reviewer to item 3 above with related discussion. While this would need to be tested in our specific setting, recent literature suggests gains on encoders are not as pronounced with scaling as they are for decoders. This is also confirmed by the fact that our best performing models were not the biggest ones. We also highlight that our decision in terms of which models to fine-tune was in part motivated by the amount of compute we had available for this project.
>
> **References**
>
> [1] Wang Y, Le H, Gotmare AD, Bui ND, Li J, Hoi SC. Codet5+: Open code large language models for code understanding and generation. arXiv preprint arXiv:2305.07922. 2023 May 13.

---

> > ### Author Response · Authors · 2023-11-19
> > **Kind reminder**
> >
> > We thank the reviewer for their thoughtful comments and detailed assessment. As we get closer to the end of the discussion period, kindly let us know if we were able to clarify all of your concerns or if there is anything else we could address.

---

### Official Review · Reviewer_MZBA · 2023-11-02

**Soundness:** 2 fair
**Presentation:** 2 fair
**Contribution:** 2 fair
**Rating:** 3
**Confidence:** 4

**Summary:**

The paper proposes a dataset called $M^2CRB$ where docstring and corresponding functions are used as paired data to construct a search dataset. The dataset includes docstrings in Spanish, Portuguese, German, and French. The paper proposes a training recipe that enables search over unseen language pairs. The paper reports the effects of different design choices and the performance of various fine-tuning schemes.

**Strengths:**

A new dataset for multilingual code retrieval task.

**Weaknesses:**

The paper proposed a dataset that is created automatically. The training and evaluations are not motivated well. In the whole paper, every experiment performed is not justified. The main questions addressed in the paper are somewhat known. In my opinion, the paper does not meet the bar of ICLR. There is no scientific or technical contribution. I couldn't perceive the value and need of the dataset.

**Questions:**

- If we need multilingual docstring, isn't it possible to use NMT models to translate the docstring available in English? Some experiments in the Appendix use Google translation API which should be part of the main body and discussion. This way the value of the dataset could be better demonstrated.
- "Moreover, the search setting is less compute-intensive relative to common language models, rendering experimentation more accessible" - Is it a good reason to study code search tasks?
- The research question related to unseen languages is not clear. From the literature, we know that multilingual LMs learn to map text or code in a shared embedding space that enables them to perform on unseen language pairs. What new this paper is trying to show or prove?
- "Contrary to what is usually observed in more standard evaluation conditions, in this multilingual setting, scaling up model size will not necessarily improve performance." - Why?
- Why does a retriever model need to be trained to generate? When we are in the era of large language models that are doing great in code generation, why do we want to train CodeBERT-like models for generation?
- In 2-3 lines, please outline what new findings this paper presents.

---

> ### Author Response · Authors · 2023-11-16
>
> We thank the reviewer for their assessment and for seeking to ensure our contribution is relevant, which we believe will help improve our work. Please see below for detailed responses and comments.
>
> **1. training and evaluations are not motivated well.**
>
> **Response:** We agree with your other comment, *i.e.*, “*we know that multilingual LMs learn to map text or code in a shared embedding space that enables them to perform on unseen language pairs*”. However, the embedding spaces learned by LMs do not immediately lead to embeddings which can be used to query code for retrieval applications. One of the motivations of this work is to show that we can adapt the representations learned by LMs and make them much more effective for cross-lingual search.
>
> We further clarify that evaluations follow the standard information retrieval setting where, given a query, one seeks $k$ relevant entries from a database. That’s our target application as discussed in the text. We discuss motivations further in item 5 below. As per training, we used contrastive learning approaches to yield a semantic embedding that is language agnostic so as to enable multi-linguality.
>
> While we believe the summary above is written down somewhat clearly in the text, we would be happy to modify any part of the content that the reviewer believes is confusing or misleading.
>
> **2. The main questions addressed in the paper are somewhat known.**
>
> **Response:** The main question we seek to address is as follows: **Can a contrastively trained embedding enable retrieval for language combinations only indirectly seen during training?** As we argued in the text, if yes, then training of this kind of model can be carried out without paired data from all languages in the set of languages of interest. **To our knowledge, ours is the first work showing evidence of that being the case.**
>
> **3. I couldn't perceive the value and need of the dataset. If we need multilingual docstring,...**
>
> **Response:** That’s actually the main motivating point of the line of work we carry out in this paper: **translation steps add overhead and should be removed**. This translation overhead (be it machine or human translation) mainly affects non-English-speaking communities of developers or developers required to write documentation in local language. We thus make available both data and efficient training strategies that help bypass undesirable intermediate translation steps.
>
> We also highlight that existing machine translators are trained in natural language, and don’t necessarily work well on docstrings and code comments which embed jargon, variable names, and other code specific information. Machine translation might also be unreliable in low-resource languages.
>
> **4. …isn't it possible to use NMT models to translate the docstring available in English?**
>
> **Response:** Please refer to Table 10 in the appendix where we compare direct search with models that use an external translator of docstrings. We see that multilingual models perform better than *translate+search* using an English-only model. We also see that multilingual models can even further benefit from translating docstrings into English, so even if translation is an option, multilingual training is beneficial.
>
>
> **5. good reason to study code search tasks?**
>
> **Response:** The main motivation we considered for doing code search as opposed to generation is to address situations where **users wish to obtain pieces of code from a known and tested codebase**. There are many situations where the user desires retrieval results and in the case of retrieval augmented generation (RAG) models, our approach can serve as the underlying representation used to perform the retrieval step as well as a generator.  One can also consider the situation of proprietary codebases that are not in the training set of large code LMs. In those cases, generative models wouldn’t be able to generate code that makes sense in the context of an API or library it never had access to and knows nothing about.
>
> We additionally highlight that code search is an **extremely active research field** with several papers coming out every week, especially so given the recent trend in combining retrievers with decoders to perform **retrieval augmented generation (RAG)**.
>
> Finally, as noted by the reviewer, we also highlighted in the paper that retrieval applications are typically less expensive in terms of compute resources requirements, and might be more accessible as a research topic to groups with compute constraints.

---

> > ### Author Response · Authors · 2023-11-16
> >
> > **6. The research question related to unseen languages is not clear. From the literature, we know that multilingual LMs learn to map text or code in a shared embedding space that enables them to perform on unseen language pairs. What new this paper is trying to show or prove?**
> >
> > **Response:** We kindly clarify that we address code search from text, specifically focusing on enabling multi-linguality for contrastive models. While results in multilingual language modeling are indeed interesting and relevant, as discussed above, **we care about situations where generation is not an option** and search from a given codebase is required. To put it differently, multilingual search and generation are orthogonal problems, and thus results on one will not necessarily transfer to the other.
> >
> > **7. "Contrary to what is usually observed in more standard evaluation conditions, in this multilingual setting, scaling up model size will not necessarily improve performance." - Why?**
> >
> > **Response:** As discussed in the text, we believe this to be due to the distribution shift we observe in our evaluations. Most existing work on the definition of scaling laws observed monotonic improvements with capacity specifically for in-distribution evaluation. Our evaluation on the other hand, induces a **severe distribution shift** since we evaluate models on language combinations **completely absent from the training data**. Our results thus suggest an in-distribution vs. out-distribution tradeoff controlled via model capacity.
> >
> > **8. Why does a retriever model need to be trained to generate? When we are in the era of large language models that are doing great in code generation, why do we want to train CodeBERT-like models for generation?**
> >
> > **Response:**  We kindly clarify that we train a broad range of model classes and some of them have decoders that can be used to generate (not CodeBERT). We highlight that our initial motivation for this project was to harness the ability of LLMs and enable them to further retrieve, but included encoder-only models such as CodeBERT since those still excel on search settings.
> >
> > To be more clear, we only retrieve with encoders, and CodeBERT-like models are only used for retrieval. However, we additionally trained and evaluated encoder-decoder models (T5-like)  and decoder-only models (GPT-like), and in those cases we used the available decoder to generate code while carrying out retrieval with the encoder. This is to show that one can simultaneously generate an answer given a query and search a codebase so the user can get both outputs, all with the same model. Our results showed that this multi-task architecture performs almost as well as encoder-only models for search, which is convenient for practical applications.
> >
> > **9. In 2-3 lines, please outline what new findings this paper presents.**
> >
> > **Response:** While the two datasets we open-source are central contributions, our paper reports several findings the research community can benefit from, as outlined below and discussed in the manuscript:
> >
> > 1- We show evidence that multilingual contrastively trained encoders generalize to language combinations only indirectly observed during training.
> >
> > 2- We show that encoder-decoder and decoder-only LM architectures can effectively be trained to perform both search and generation, and no performance tradeoffs are observed in this case.
> >
> > 3- We show that multi-linguality brings in specific challenges and design choices must be made with this setting in mind. For instance, dual-encoder architectures, state-of-the-art for text retrieval, are outperformed by single encoders in this situation.
> >
> > 4- We show that multilingual LLMs can be used for evaluating semantic match between docstrings and code snippets. Interestingly, the judgments of GPT-3.5-turbo correlate with `BERTScore`, and both are predictive of search performance.

---

> > > ### Author Response · Authors · 2023-11-19
> > > **Kind reminder**
> > >
> > > Thank you again for reviewing our paper and providing useful feedback. Since we are getting close to the end of the discussion period, we would like to kindly ask the reviewer whether our rebuttal fully answered their questions about the motivation of our work and usefulness of the datasets we introduced and experiments we reported. Please let us know if there are remaining concerns we should address, and if not, kindly consider all the information added during the discussion period for the evaluation of our paper.

---

> > > ### Comment · Reviewer_MZBA · 2023-11-22
> > > **Thank you for your answers.**
> > >
> > > Thank you for taking time to answer my questions. However, I am not convinced that the work is significant. The paper highlights that it shows evidence that multilingual contrastively trained encoders generalize to language combinations only indirectly observed during training. This is perhaps not true. For example, UniXCoder was trained via contrastive learning on 6 languages, and it shown it work well on zero-shot code-to-code search tasks. Source code across programming languages often share a high lexical overlapping that can play the anchor role to learn multilingual alignment. At least that is my expectation.
> > >
> > > Overall the paper does not meet the bar for ICLR in my opinion. Therefore, I am retaining my score.

---

> > > > ### Author Response · Authors · 2023-11-22
> > > >
> > > > We thank the reviewer for checking our responses and pointing us to UniXCoder, which we agree is very relevant and we now cite. However, we fail to see how their results in any way affect the impact of our contribution. UniXCoder leveraged feature engineering and fed the model with flattened AST parsed code to induce invariance, while we proposed an approach to learn invariant representations directly from tokens.
> > > >
> > > > Both approaches are valid and can be combined, and both target the same goal: obtain language invariant representations of text and code. One can indeed manually devise representations that satisfy invariance such as the flattened AST they used in UniXCoder, but that only works for languages one knows how to parse. Our approach seeks the same but is more general in that it doesn't assume access to a parser to train. In other words, the two approaches show different things: we show invariance can be obtained by combining multiple paired data sources and contrasting pairwise. They show that AST parsers can be used to help representing code in a language-agnostic manner.
> > > >
> > > > In any case, even if the reviewer believes that that specific component of our work is of less importance given that prior work had showed other approaches with a similar objective, we highlight that our paper has several other contributions that were highlighted in our original response as per the reviewer's request, and those add independent value.

---

### Author Response · Authors · 2023-11-16

We thank all reviewers for the detailed assessment of our dataset and the empirical evaluation we performed on top of it. We are glad they found our evaluation to be rigorous and extensive (Reviewer z3xc), that they appreciated that our experiments cover different models, sizes, and languages extensively  (Reviewer mHNB), that our proposal is novel and re-usable in other settings such as text-conditional code modeling (Reviewer QVKG), and that our paper clearly presents the data filtering pipeline, training methodology, and results, making those easy to follow and understand  (Reviewer z3xc). Reviewer MHNB further highlights that realistic evaluation datasets are needed, and we believe our contributions help to fill this gap.

We carefully addressed all weaknesses and questions raised by the reviewers in comments to each review, and will post an updated version of the manuscript once the discussion phase ends, including all changes required during that phase.

---

### Author Response · Authors · 2023-11-20
**Kind reminder to participate in the discussion**

Dear reviewers,

Thank you again for volunteering time to review our manuscript and helping to improve it. Since the discussion period nears its end, we would like to kindly invite the reviewers to check our responses.

In particular, we really hope that the most negative reviewers, **MZBA** and **z3xc**, will consider reassessing the importance and quality of our work based on our individual responses to them and some of the comments from other reviewers. We summarize our comments addressing their main concerns in the following:

- **MZBA**: While results in language modelling in multilingual settings are indeed impressive, they don't solve retrieval, and retrieval applications are extremely important in situations where generated code cannot be trusted (*e.g.*, the user requires code that makes sense in the context of a proprietary closed-source API or library). Moreover, retrieval augmented generation approaches (RAG) are now the dominant language modelling framework, and those require a good retriever to be available.

- **z3xc**: Our evaluation covered a large number of different methods and models classes. For instance, encoder-only, dual encoders, encoder-decoder, and decoder-only models were all considered. Moreover, we covered the 60M-360M parameters range, which is **rather typical for sequence-level representation learning**. For example, the extremely popular `sentence-transformers/all-MiniLM-L12-v2` encoder has only 33M parameters. Another recently released encoder, `bigcode/starencoder`, has 125M parameters. The range we chose is very well aligned with common practice in the code search setting.

We look forward to hearing back and will be happy to provide further clarifications if needed.

Kind regards.

---

### Meta-Review · Area_Chair_9CLm · 2023-12-14

**Metareview:**

First, this work introduces a code dataset comprising natural language query--code snippet pairs. The natural languages included are Spanish, Portuguese, German, and French, and the programming languages are Python, Java, and JavaScript. It is filtered, cleaned, and annotated from GitHub data. In total, it contains 7700+ pairs.

Then the authors seek to test the hypothesis: "if a model can map from English to code, and from other natural languages to English, then the model can directly map from those non-English languages into code." To examine this, they create an additional dataset featuring English-code snippet pairs.

Finally, they compile a training corpus that includes 1) English-code snippet pairs and 2) translation pairs between English and non-english languages. This corpus is used to train code models, yielding findings and interesting results.

The scope of this work is quite broad, in other others not converged. Individually, each of the aforementioned contributions does not offer significant value to the field, as the dataset being small-scale and the subsequent study failing to yield novel insights or perspectives. When these items are combined, they unfortunately do not coalesce into a coherent narrative, nor do they significantly enhance the technical contributions or insights of the research.

**Justification For Why Not Higher Score:**

This submission receives on average negative ratings (3, 5, 6, 6), as the reviewers have found it challenging to pinpoint contributions that reach the standard of ICLR or offer meaningful insights and perspectives that the community can draw inspiration from or build upon.

**Justification For Why Not Lower Score:**

n/a

---

### Decision · Program_Chairs · 2024-01-16

Reject